# Drug-Eluting Embolic Loaded with Tyrosine Kinase Inhibitor Targeted Therapies for Transarterial Chemoembolization in a VX2 Model

**DOI:** 10.3390/cancers15123236

**Published:** 2023-06-18

**Authors:** Nadine Abi-Jaoudeh, Ben Sadeghi, Hanna Javan, Jim Na, Graham Beaton, Fabio Tucci, Satheesh Ravula, David K. Imagawa

**Affiliations:** 1Department of Radiological Sciences, University of California Irvine, Orange, CA 92697, USA; bsadeghi@hs.uci.edu (B.S.); hanna_javan@yahoo.com (H.J.); 2Cullgen, Inc., San Diego, CA 92130, USA; drjamesna@gmail.com; 3Epigen Biosciences, San Diego, CA 92121, USA; gbepigen@gmail.com (G.B.); ftucci@epigenbiosciences.com (F.T.); sravula@epigenbiosciences.com (S.R.); 4Department of Surgery, University of California Irvine, Orange, CA 92697, USA; dkimagaw@hs.uci.edu

**Keywords:** transarterial chemoembolization (TACE), drug-eluting embolics (DEEs), Sorafenib, Regorafenib, VX2 model

## Abstract

**Simple Summary:**

Embolization shuts down vessels and cuts off the supply of blood to liver tumors. Using small beads, drugs can be delivered locally. Thus far, the drugs loaded onto the beads have not been tailored to liver tumors because the physical properties of efficacious drugs precluded it. Our patented formulation enables targeted therapies to be loaded onto the beads. This study aims to examine the safety and feasibility of VX2 rabbit tumor model embolization using beads loaded with the targeted therapies. The study showed that embolization was possible. The targeted therapies eluted locally and did not escape into systemic circulation.

**Abstract:**

Drug-eluting embolic transarterial chemoembolization (DEE-TACE) improves the overall survival of hepatocellular carcinoma (HCC), but the agents used are not tailored to HCC. Our patented liposomal formulation enables the loading and elution of targeted therapies onto DEEs. This study aimed to establish the safety, feasibility, and pharmacokinetics of sorafenib or regorafenib DEE-TACE in a VX2 model. DEE-TACE was performed in VX2 hepatic tumors in a selective manner until stasis using liposomal sorafenib- or regorafenib-loaded DEEs. The animals were euthanized at 1, 24, and 72 h timepoints post embolization. Blood samples were taken for pharmacokinetics at 5 and 20 min and at 1, 24, and 72 h. Measurements of sorafenib or regorafenib were performed in all tissue samples on explanted hepatic tissue using the same mass spectrometry method. Histopathological examinations were carried out on tumor tissues and non-embolized hepatic specimens. DEE-TACE was performed on 23 rabbits. The plasma concentrations of sorafenib and regorafenib were statistically significantly several folds lower than the embolized liver at all examined timepoints. This study demonstrates the feasibility of loading sorafenib or regorafenib onto commercially available DEEs for use in TACE. The drugs eluted locally without release into systemic circulation.

## 1. Introduction

Trans-arterial chemoembolization (TACE) is the standard of care for intermediate hepatocellular carcinoma (HCC) according to several guidelines, including the Barcelona Clinic Liver Cancer (BCLC) and the National Comprehensive Cancer Network (NCCN) guidelines [1,2]. As there is no standardized TACE technique, outcomes are heterogenous [3,4,5]. Moreover, the addition of chemotherapeutic agents to embolization has been questioned as several randomized controlled trials did not show a benefit to using TACE vs. trans-arterial embolization (TAE) alone [6,7,8,9,10,11,12]. Several critiques were leveled at the trials, but perhaps the most relevant critique is that the agents used in TACE are not tailored toward HCC [13,14]. In fact, doxorubicin is the most widely used agent in TACE, though it has been shown in several large randomized trials to be ineffective in HCC [11,14,15]. Choosing agents that are tailored to HCC may help improve TACE outcomes. Several angiogenic agents have been demonstrated to be very effective against HCC, but their physical properties preclude them from being loadable and delivered by commercially available embolic agents [16,17]. To load a therapeutic agent onto a drug-eluting embolic (DEE), the agent must be hydrophilic and water-soluble. Embolic agents such as PVA-based imageable microspheres are being developed with improved tissue adhesion and the loading capabilities of water-soluble anionic drugs, such as 5-fluorouracil, but they cannot be loaded with lipophilic drugs [18]. Sorafenib and regorafenib are lipophilic and neutral drugs that are efficacious in HCC, and up until 2020, they were the first- and second-line standard-of-care systemic therapies for advanced HCC [19,20]. Due to their properties, neither drug can be loaded onto DEE for local delivery, which would avoid many of the systemic side effects experienced by patients. Our group patented a platform to enable the loading of commercially available lipophilic and neutral drugs on any embolic, including commercially available DEEs. The process entails loading the desired therapeutic agent on cationic liposomes that are, in turn, loaded onto a negatively charged DEE. Preliminary ex vivo testing demonstrated that the liposomal formulations of sorafenib and regorafenib can be loaded with therapeutic agents on commercially available DEEs and are eluted from the embolic material in a predictable and reproducible manner. In this study, we present the feasibility and pharmacokinetic profile of TACE using sorafenib- or regorafenib-loaded DEEs in a VX2 rabbit tumor model.

## 2. Materials and Methods

The animal studies were performed under the auspices of the institutional animal care use committee (AUP-21-005-(17-118)). Adult New Zealand white male and female rabbits weighing between 3.6 and 4.1 kg were utilized in this study. 

### 2.1. Tumor Model Creation

The tumor model has already been described and validated [21,22,23]. In brief, a VX-2 cell line was propagated in a cell culture and injected into the hind limb muscles of donor rabbits. The VX-2 cell line was procured from the National Institutes of Health Center for Interventional Oncology Laboratory. Once the tumors had grown, they were excised, and the animals were humanely euthanized. The tumors were sectioned and implanted into the livers of recipient rabbits via a sub-xyphoid midline incision. The tumor implantation was performed under sterile conditions, with the rabbits maintained under general anesthesia using isoflurane 0.75–4% after induction via 50 mg/kg of ketamine administered intra-muscularly (IM) and 5 mg/Kg of Xylazine administered subcutaneously. Prophylactic antibiotics were provided (5 mg/Kg of Enrofloxacin, administered intramuscularly, BID). A midline sub-xyphoid incision was used to expose the hepatic parenchyma. The tumor cells were implanted into the left lobe of the liver via a 24 Ga angiocatheter. Care was taken not to spread cells in the abdominal cavity. The abdominal incision was then sutured, and the animal recovered. 

The rabbits were monitored frequently for any signs of distress or discomfort post tumor implantation. The tumors were monitored via ultrasound once per week. Once the tumors reached 2.5–3.5 cm, the embolization procedure was performed. 

### 2.2. Sorafenib/Regorafenib Liposomal Formulation and Loading onto Drug-Eluting Embolic (DEE)

Liposomal formulations have been developed for drug delivery in clinical settings [24]. Reagents were used to prepare a cationic liposomal formulation of sorafenib and regorafenib that could be loaded onto commercially available, negatively charged embolics (DEEs). This process was awarded a US patent protection as a platform for drug delivery into solid organ tumors in 2023. Briefly a tyrosine kinase inhibitor, such as sorafenib para-toluene sulfonate salt, 1,2-dioleoyl-3- trimethylammonium-propane chloride salt (DOTAP), or 1,2-dioleoyl-sn- glycero-3-phosphocholine (DOPC), was dissolved in dehydrated ethanol. The solution was added dropwise to a 10% trehalose solution while stirring. The resulting emulsion was stirred at room temperature for an additional 5 min and filtered through a manual liposomal extruder using a 200 nm polycarbonate membrane. The unilamellar liposomal solution was lyophilized overnight. The liposomal formulated tyrosine kinase inhibitor was mixed with distilled water. Drug-eluting embolics were drained out of the fluid, leaving 2 mL of beads which were added to the liposomal formulated sorafenib or regorafenib. The loading and elution profiles were examined in vitro. Although beyond the scope of this manuscript, the range of loading efficiencies, which depended on conditions tested, varied between 41 and 51.9%. The elution profiles are shown in Figure 1 and Figure 2. At 320 min, 82.6% of sorafenib and 74.2% of regorafenib had eluted off of the beads. 

The liposomal formulated sorafenib and regorafenib, in freeze-dried form, were prepared at two doses of 18 mg or 36 mg (Epigen laboratory, San Diego, CA, USA) and kept frozen until two days prior to the embolization procedure. In the first 12 rabbits, two sorafenib concentrations were tested, 4.5 and 9 mg/mL, with the animals sacrificed at 1 h, 24 h, and 72 h post embolization. This was achieved by mixing 2 mL of distilled water with either an 18 mg or 36 mg vial of liposomal sorafenib vial to generate either 9 or 18 mg/mL of liposomal sorafenib concentration with an appropriate viscosity for injection through a syringe. This mixture was added to a 2 mL vial of DEE (LC beads, 100–300 um, Boston Scientific, Minnetonka, MN, USA). The final concentration was either 4.5 mg/mL or 9 mg/mL of DEE-loaded sorafenib. An additional 4 animals were tested using the 4.5 mg/mL concentration, with 2 animals sacrificed at 24 h and 2 at 72 h. Two animals were tested with bland beads, and five animals were embolized with regorafenib-loaded beads; two were sacrificed at 24 h, and three were sacrificed at 72 h. For the regorafenib group, 18 mg of liposomal regorafenib was reconstituted in 2 mL of distilled water to generate 9 mg/mL of liposomal regorafenib, which was added to a 2 mL vial of DEE (LC beads, 100–300 um, Boston Scientific USA) for a final concentration of 4.5 mg/mL of regorafenib-loaded DEEs. The DEEs were left to stand at room temperature for 30 min prior to TACE. 

### 2.3. Trans-Arterial Chemoembolization (TACE) Procedure

The embolization procedure has been described previously [23,25,26,27]. The procedures were performed under aseptic conditions with prophylactic antibiotics. The animals were maintained under general anesthesia. A C-arm unit (Pulsera BV, Philips Medical Systems Inc., Best, The Netherlands) was used for angiography. The femoral artery was accessed through a surgical cut-down and catheterized with a 3 F vascular sheath, after which a 2.4 F microcatheter (Progreat, Terumo, Irvine, CA, USA) and microwire (Transcend 0.018”, Stryker Inc., Kalamazoo, MI, USA) were advanced into the common hepatic artery. Angiography was performed. The tumor was visualized as a region of hypervascular blush in the liver. The hepatic artery supplying the tumor was catheterized as selectively as possible and embolized using the DEE loaded with liposomal sorafenib or regorafenib under fluoroscopic visualization until vascular stasis was achieved. Post TACE, the femoral artery was ligated. Post procedure, the animals were monitored daily for signs of discomfort. Doses of Buprenorphine ranging from 0.01 to 0.05 mg/Kg and a 5 mg/Kg dose of Enrofloxacin were administered intramuscularly or subcutaneously if needed. 

### 2.4. Pharmacokinetic Analysis and Pathology

For the measurement of the systemic agents’ levels, blood samples were collected at 5 and 20 min and 1 h, 24 h, and 72 h post TACE, prior to the euthanasia of the animals. Within 10 min of the sacrifice, a rabbit necropsy was performed, and the liver was harvested for tissue analysis. Post sacrifice, the treated tumors were extracted and divided in half. In addition, representative 2 cm^3^ samples of non-tumorous liver parenchyma—from the left hepatic lobe and the right hepatic lobe—were also procured from each rabbit. The specimens were frozen at −80 °C until histopathological analysis. Measurements of sorafenib or regorafenib were performed in all tissue samples in each animal to limit sampling errors using the same mass spectrometry method. The bioanalysis was conducted at Quintara Discovery. The liver samples were first homogenized in two volumes of ice-cold water. An aliquot of 20 µL of the plasma and liver homogenate samples was treated with 100 µL of methanol/acetonitrile (5:95) containing an internal standard (Verapamil). The mixture was vortexed on a shaker for 15 min and subsequently centrifuged at 4000 rpm for 15 min. An aliquot of 70 µL of the supernatant was mixed with 70 µL for the injection into the LC/MS/MS. Calibration standards and quality control samples were prepared by spiking the test compound into the blank matrix and then processed with the unknown samples. 

In addition, histopathological examinations of the tumor and non-tumor tissues were carried out. The tissues were cut and prepared into five paraffin blocks, each 500 mm thick. From each block, 5 mm thick slices were taken with a microtome, and each slice was stained with hematoxylin and eosin (H&E). The H&E slides were evaluated under low magnification (25×) for a general assessment of the distribution of the percentage of necrosis or damage (if present), the hepatocellular cytoplasmic degeneration and the attributed zone, followed by higher-magnification assessments (100× and 400×). TUNEL and Cleaved Caspase-3 staining were used to quantify the extent of cell death in the samples at various timepoints.

### 2.5. Statistical Analysis

Sorafenib, regorafenib, and laboratory values were provided using descriptive statistics as means and medians ± standard deviations for each time point. The liquid chromatography–tandem mass spectrometry analysis was performed by Quintara Discovery laboratories.

## 3. Results

Experiments were performed on 23 rabbits. The rabbits were divided into several groups according to the drug administered and the timepoints. A total of 4 rabbits were sacrificed at 1 h, 9 rabbits at 24 h, and 10 rabbits at 72 h. Of note, two rabbits from the 24 h group were euthanized prematurely for humane reasons, leaving seven rabbits in the 24 h group with pharmacokinetics and pathology at the 24 h time point. The rabbits were divided as follows: 16 sorafenib, 2 bland, and 5 regorafenib. In the sorafenib group, 10 rabbits received the lowest concentration of 4.5 mg/mL, and 6 received the higher concentration of 9 mg/mL. In the sorafenib group, four were sacrificed at 1 h post TACE (two in the 4.5 mg/mL group and two in the 9 mg/mL group). At 24 h, two rabbits were sacrificed for humane reasons, leaving two at the 9 mg/mL concentration and two at the 4.5 mg/mL concentration of sorafenib-loaded DEE. Six rabbits in the sorafenib group were sacrificed at 72 h, with four receiving the 4.5 mg/mL sorafenib-loaded DEE and two rabbits assigned to the 9 mg/mL concentration. Two rabbits in the bland group were sacrificed at 24 h (n = 1) and 72 h (n = 1). In the regorafenib group, two rabbits were sacrificed at 24 h, and three rabbits were sacrificed at 72 h. 

The sorafenib and regorafenib loading was successful in all cases except one case in the regorafenib group, which was eliminated from analysis. Tumors extended across the entire left lobe in four animals and were present on the right side in three animals. Catheterization and embolization were achieved in all animals (Figure 3). Embolization was performed until stasis. As much as possible, every attempt was made to completely deliver a dose of 1 mg/kg. There were no complications. 

### 3.1. Sorafenib Group

In the sorafenib group, the median volume injected was 1 mL ± 0.21 mL (average 0.99 mL). The median dose administered was 4.5 mg ± 2.31 (average dose 6.42 mg). The plasma concentration of sorafenib was statistically significantly lower than the embolized tissue concentrations of sorafenib at all timepoints. At 1 h, the peak plasma concentration of sorafenib was at its highest but was still >10× lower than the embolized liver sorafenib concentration value. The plasma concentration decreased at 72 h and was over 90-fold lower than the embolized liver concentrations (Table 1 and Table 2). At 1 and 24 h in the sorafenib group, there was no statistical difference between the concentrations of sorafenib in the embolized and non-embolized areas, although there was a tendency to have higher concentrations of sorafenib in the embolized areas. This was not found at 72 h in the sorafenib group, in which the difference between the embolized and non-embolized areas were higher concentrations in the embolized areas. Of note, the non-embolized areas also had statistically different concentrations compared to plasma at all timepoints for the sorafenib group. 

### 3.2. Regorafenib Group

In the regorafenib group, the median volume of DEE-TACE was 0.7 mL ± 0.61 (average 1.1 mL). The median dose delivered in the regorafenib group was 3.15 mg ± 2.59 mg (average dose 7.8 mg). Regorafenib plasma concentrations peaked at 24 h (Table 2 and Table 3). The analysis was combined for the regorafenib group due to the small number (five animals total). The embolized areas had significantly statistically higher concentrations of regorafenib compared to the non-embolized areas and compared to plasma. The non-embolized area had lower concentrations than the embolized area but was still significantly higher than the plasma concentrations. 

### 3.3. Histopathology

On pathology, the beads were limited to the tumor areas in 14/23 animals. In six rabbits, beads were found in the normal left hepatic parenchyma adjacent to the tumors, and in three cases, beads were observed in the right lobe without a tumor. This is caused by reflux of the embolic material. However, in four cases, beads were observed in the portal system. It is unclear if this was due to shunting or from extensive necrosis. In seven animals, there were tumors that were not treated on the contralateral side or in another area of the left lobe. In tumors where beads were found, the percentage of tumor necrosis was 75% ± 33.2%, while in the rabbits in which no beads were found in the tumors, the percentage of necrosis was 40% ± 18% (*p* = 0.007) (Figure 4 and Figure 5). Other than necrosis, tumoral changes and moderate non-specific changes were also observed, including zone 2 and 3 hepatocyte ballooning degeneration, portal inflammation, and edema. These changes were present regardless of the timepoint at which the animal was sacrificed.

## 4. Discussion

This study demonstrated the feasibility of trans-arterial chemoembolization using sorafenib or regorafenib loaded onto commercially available embolics (DEEs) using liposomal formulation in a VX2 tumor model. The agents were delivered intra-arterially on the beads and eluted locally into the tumor with minimal escape into the systemic circulation. At all timepoints in our study, the plasma concentrations for both sorafenib and regorafenib were statistically lower than the homogenate tissue concentrations. This technique provides several advantages. Firstly, TACE has been shown to improve overall survival in the treatment of unresectable intermediate HCC [2,3,5,28]. However, local recurrence or incomplete treatment is common, especially in larger tumors [29]. Conventional TACE was initially performed using lipiodol-mixed lyophilized formulations of cytotoxic agents, followed by embolic agents [12,28]. Numerous cytotoxic agents have been used in cTACE, but the most ubiquitous combination was doxorubicin, mitomycin, and cisplatin [12,30]. Techniques to improve TACE outcomes have been attempted, including cytotoxic agents’ delivery into the tumor via DEE-TACE [27,31,32,33]. Several large, prospective randomized trials have failed to show improved response rates, PFS, or OS of DEE-TACE vs. cTACE [34,35,36]. In fact, prospective randomized trials have failed to show that DEE-TACE is better than bland trans-arterial embolization (TAE) [6,7,11]. A network meta-analysis of 17 prospective randomized trials encompassing 2330 patients found that the combination of doxorubicin, mitomycin, and gentamycin in TACE produced the best response rates, overall survival, and least adverse events [11]. Single-agent TACE did not show any benefit in terms of response rates, OS, or adverse events compared to bland TAE. However, all agents included in this meta-analysis were cytotoxic chemotherapies with minimal efficacy against HCC, unlike targeted therapies. Moreover, a survey of practices found that doxorubicin was the most commonly used agent in TACE [30,37,38]. Unfortunately, several large clinical randomized trials have demonstrated no benefit to systemic doxorubicin in HCC [14]. In fact, the latest trial combining doxorubicin with sorafenib vs. sorafenib alone failed to show any improvement in OS or PFS in the combination group but demonstrated increased adverse events [15]. In other words, the addition of doxorubicin resulted in worse outcomes for patients. In another study, 11 therapeutic agents, including doxorubicin, were tested on three HCC human cell lines [13]. The latter demonstrated a low cytotoxic effect on two HCC cell lines and no effect on one of the cells, while idarubicin was the only drug that showed cytotoxic effects on all three cell lines. In addition, doxorubicin’s cytotoxic effects are reduced in an acidic microenvironment as its ability to penetrate the cellular cytoplasm is diminished [39,40]. Embolization renders the extra-cellular tumor microenvironment hypoxic and acidic [40,41]. An acidic extra-cellular tumor microenvironment reduces immune cell infiltration and is associated with lower response rates and poorer outcomes [41,42,43,44]. The widespread use of doxorubicin in TACE, either in combination with other drugs in conventional TACE or as a monotherapy in DEE-TACE, is not justified with current evidence [13,14]. Therefore, interest in better agents for TACE has been renewed recently. Among chemotherapeutic agents that demonstrate cytotoxicity to the HCC cell lines, idarubicin was the most potent and showed activity against all three cell lines [13]. Studies exploring DEE-TACe with idarubicin have shown promising results [45,46]. However, a more promising avenue is the combination of target therapies or hypoxia agents in embolization procedures [47] as the mechanisms are complementary [48]. Indeed, another significant contributor to TACE failure is the ensuing hypoxic conditions resulting from the embolization [12,43,49]. Pre-clinical and clinical studies have demonstrated up-regulations of hypoxia-inducible factor 1a (HIF-1a) and vascular endothelial growth factor (VEGF) post embolization [26,42]. Studies have shown that patients with baseline elevated HIF-1a and VEGF had worse overall prognoses and greater increases in their serum HIF-1a and VEGF levels post TACE [44,49]. These levels did not return to normal post TACE. Moreover, some studies have shown that the increases in HIF-1a and VEGF post TACE correlate with a lack of response [42,44,50]. HIF-1a has been associated with an increase in tumor angiogenesis through the activation of VEGF and platelet-derived growth factor (PDGF) pathways [50,51,52]. HIF-1a also promotes the Warburg effect, with changes from oxidative phosphorylation to anaerobic glycolysis enabling survival in hypoxic conditions [51,52]. This metabolic transformation also reduces tumoral cell proliferation, which is a key mechanism of action for chemotherapy and radiotherapy [53]. Counteracting the hypoxia produced by TACE inspired the rationale of combining it with anti-VEGF drugs, including sorafenib, brivanib, and orantinib [48,54,55]. Several large randomized trials combining systemic sorafenib and TACE vs. TACE alone failed to meet their endpoints [56,57,58]. Several criticisms were leveled at the trials as explanations for the negative results including the scheduled DEE-TACE regimen, a short sorafenib regimen in the SPACE trial [56], a prolonged time between TACE and the initiation of sorafenib in the post-TACE trial [58], as well as progression per mRECIST/RECIST 1.1 in the TACE 2 trial [57]. More recently, the TACTICS trial randomized patients with unresectable HCC to TACE alone vs. TACE and sorafenib [59]. In this trial, the patients began sorafenib 3 weeks before TACE, which was performed on demand only. The progression-free survival endpoint was defined as untreatable via TACE tumor progression, the deterioration of liver function or vascular invasion/metastatic disease, or death in line with updated BCLC guidelines of stage progression rather than an imaging endpoint such as progression as per mRECIST [2]. The TACTICS trial found a significant improvement in PFS in the combination group vs. TACE alone (25.2 months vs. 13.5 months; *p* = 0.006). Despite these encouraging results in PFS, which correlated with OS in HCC, only half of the cohort was able to receive the planned dose of sorafenib, and none of the patients were able to be maintained at 800 mg dose due to adverse events [59]. In some trials, upwards of 30% of patients required interruptions of sorafenib, and 20% required dose reductions due to adverse events, including cachexia and hand–foot syndrome [60,61,62]. Sorafenib is associated with significant side effects leading to dose reductions, interruptions, and poor patient compliance [62]. There is a clinical benefit to combining sorafenib and TACE which can be optimized via the local delivery of sorafenib, avoiding its systemic side effects. Considering the lack of efficacy of doxorubicin-loaded DEE-TACE and the systemic efficacy of tyrosine kinase inhibitors in HCC, sorafenib-loaded DEE-TACE can have the potential to improve patient outcomes. Moreover, trials currently underway, such as Emerald 3 and others (NCT05301842, NCT05717738, NCT04814043, NCT04472767, and NCT04803994) combine a systemic tyrosine kinase inhibitor and two or more systemic immunotherapies with chemoembolization. However, a combination of numerous drugs results in cumulative adverse events and a lack of patient compliance; therefore, the local delivery of the TKI via DEE-TACE is a very appealing concept. 

Sorafenib has not been widely used in TACE because of the drug’s properties. In a previous VX2 animal study, a powder formulation of sorafenib was mixed with ethiodized oil (Lipiodol, Guerbet) and administered in the tumor-supplying hepatic artery until stasis was achieved [25]. The mean tumor sorafenib levels were 3.53 mg/mL and normal liver parenchyma sorafenib levels were 0.75 mg/mL, while the serum levels of sorafenib were 58.58 mg [25]. The mean tissue concentrations and circulating levels of sorafenib peaked at 30 min. The peak concentration of sorafenib of 139.7 ng/mL ± 107.3 ng/mL was reached at one hour in our study; however, 20 min measurements were obtained instead of 30 min. In their study, the preparation and formulation for sorafenib delivery during TACE showed no advantages compared to intravenous administration as the systemic concentrations were higher than the local tumoral and hepatic concentrations [25]. In our study, the liposomal formulation enabled the delivery of the sorafenib into the tumor, with concentrations exceeding serum levels by >10X-90 fold, depending on the time points. This was made possible by loading sorafenib onto DEE as opposed to the intra-arterial administration of the powder formulation of sorafenib with lipiodol. Histological examinations of specimens at various timepoints in their study demonstrated nonspecific hepatocyte ballooning degeneration in the first 3 days, which resolved by the seventh day [25]. In our study, the histological findings included necrosis, and most changes remained unchanged regardless of the timepoint. It is of interest that in our study, the percentage of tumor necrosis was higher when beads were found in the tumor vs. when no beads were observed. Of note, in Gaba’s study, the administration of the sorafenib/lipiodol was not followed by an embolic material. The mixture itself was administered until vascular stasis. In cTACE studies, it has been demonstrated that without an embolic agent, there is a systemic escape of the cytotoxic agents administered with lipiodol [28,63,64]. It is unclear whether the high systemic concentrations could have been lowered through the use of an embolic material. Regardless, DEE-TACE has been associated with lower systemic concentrations of drugs, as opposed to cTACE [32,33]. However, for loading onto DEEs, drugs must be hydrophilic and have a positive charge. Sorafenib is not hydrophilic and is neutral. Attempts to use sorafenib in DEE-TACE required new, specialized beads [65,66]. In fact, sorafenib- and 2,3,5-triiodobenzoic acid (TIBA)-loaded polylactic-co-glycolic acid (PLGA) microspheres have been developed [66]. These microspheres are much smaller than what is currently used in embolization, and the risk of shunting is unclear in humans as their size correlates with radioembolization beads. Moreover, the encapsulation efficiency of sorafenib was, at best, 58% ± 0.79%, and the sorafenib content was only 5.11% [66]. The serum sorafenib concentrations were examined after oral and intra-arterial (IA) administration in that study. The AUC value was significantly higher for the oral group vs. [66]. In the IA group, however, this difference disappeared when normalized for the dose. It is of note that the oral dose was 10 mg/kg vs. the 1 mg/kg of the IA dose in that study [66]. Although this current study did not have an oral administration group, our formulation enables the use of commercially available DEEs with statistically significantly lower serum concentrations of the agents. The liposomal formulation enables the loading of other targeted therapies, such as regorafenib or Lenvatinib. In fact, the same pattern of local elution was seen with regorafenib DEE TACE. 

This study has several limitations, including the small number of animals. The VX2 model is a good embolic model that has been used in previous preclinical TACE studies, but it is not an HCC cell line and therefore, the efficacy of sorafenib on squamous cells is not documented. The VX2 tumor model grows rapidly and can become spontaneously necrotic. Rabbit hepatic vessels are small, and stasis is quickly achieved with a greater potential of reflux of DEE. Moreover, the cessation of delivery may not result in the delivery of the entire sorafenib dose. Finally, the intra-hepatic sorafenib dose was empirically determined, although it was based on preclinical (rabbits) and clinical (human) studies of effective serum concentrations.

## 5. Conclusions

In conclusion, our study demonstrated that the liposomal formulation of targeted therapies can be loaded onto commercially available DEEs and used in TACE. The local concentrations were statistically significantly higher than the serum concentrations of both sorafenib and regorafenib at all timepoints.

## Figures and Tables

**Figure 1 cancers-15-03236-f001:**
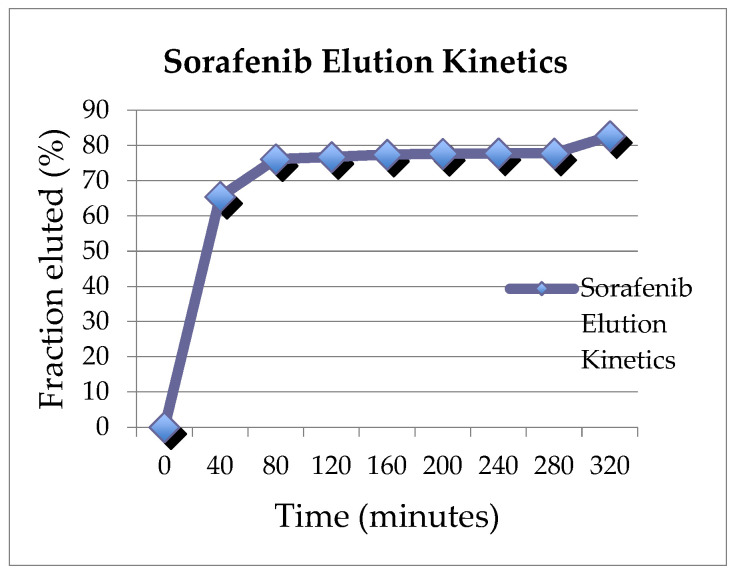
Elution profile of sorafenib.

**Figure 2 cancers-15-03236-f002:**
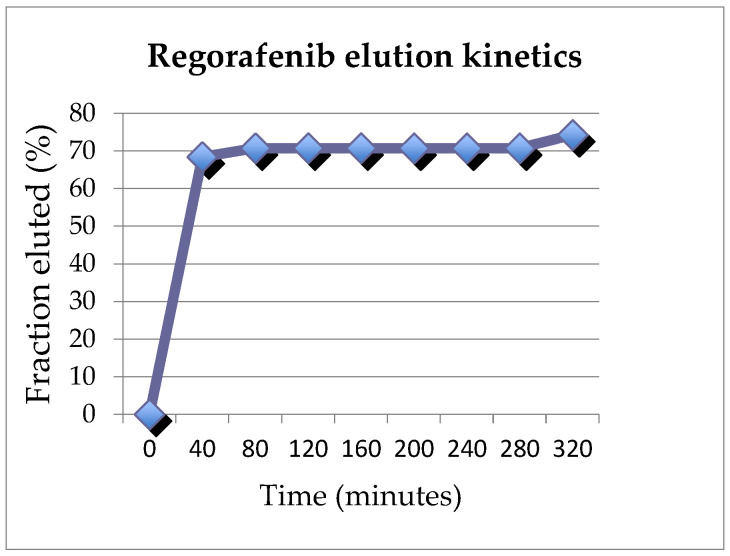
Elution profile of regorafenib.

**Figure 3 cancers-15-03236-f003:**
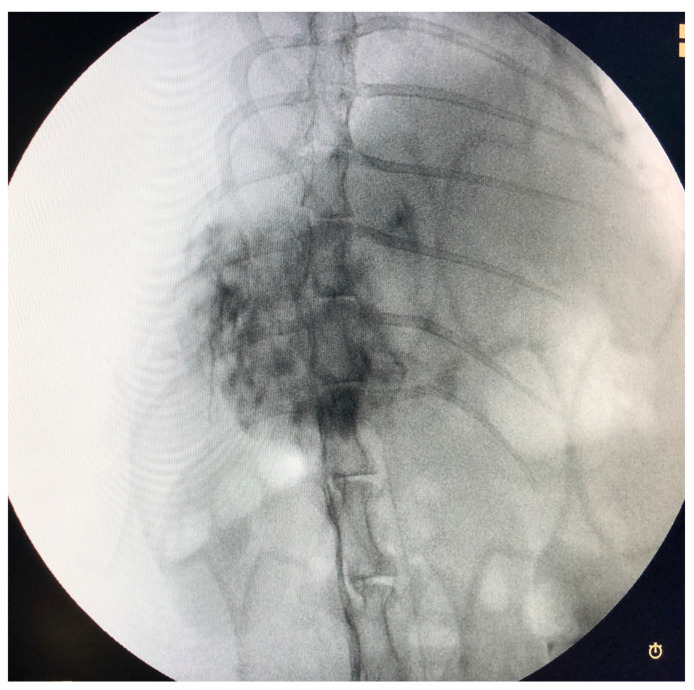
Delayed phase angiography from hepatic artery demonstrating the tumor, which can be seen as the hypervascularity.

**Figure 4 cancers-15-03236-f004:**
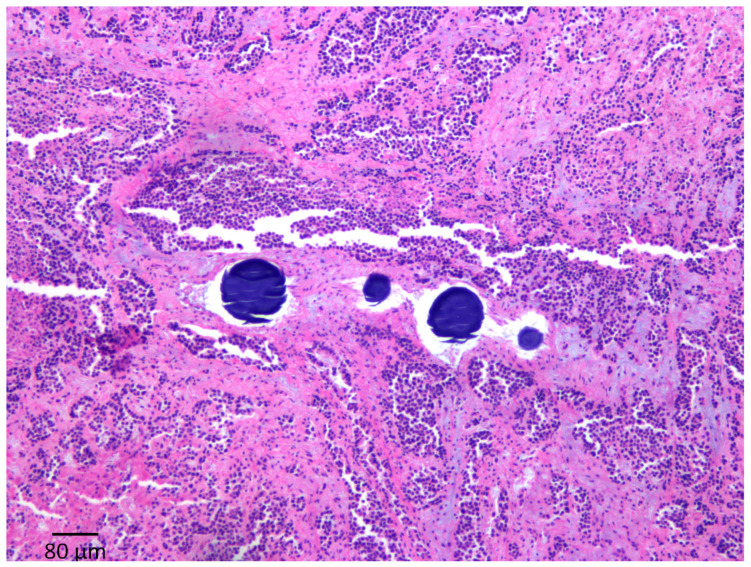
H&E stain at 100× magnification showing coagulative necrosis of the tumor associated with beads observed in the arterial lumen.

**Figure 5 cancers-15-03236-f005:**
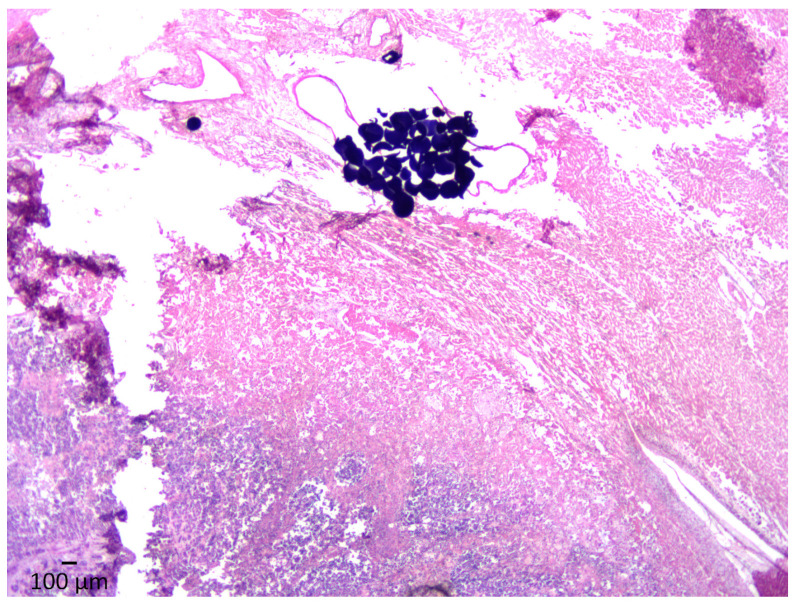
H&E stain at 25× magnification demonstrating a clump of DEEs with massive lytic and coagulative necrosis of the neoplastic tissue and adjacent hepatic parenchyma.

**Table 1 cancers-15-03236-t001:** Averages and standard deviations of serum concentrations and embolized livers as well as non-embolized liver homogenates of sorafenib concentrations.

	Sorafenib Plasma Concentration ng/mL	*p*-Value of Embolized Area vs. Plasma	Sorafenib Liver Homogenate of Embolized Area ng/mL	*p*-Value of Embolized vs. Non-Embolized Area	Sorafenib Liver Homogenate of Non-Embolized Area ng/mL	*p*-Value of Non-Embolized Area vs. Plasma
1 h	139.7 (107.3)	0.003	1534.5 (1249.4)	0.28	1080.4 (1343.9)	0.03
24 h	63.5 (37.7)	0.005	731.8 (528.5)	0.07	306.8 (290.7)	0.03
72 h	13.2 (8.6)	0.05	1233.4 (1326.6)	0.007	98.9 (56.5)	0.006

**Table 2 cancers-15-03236-t002:** Average (standard deviation) serum concentrations of sorafenib and regorafenib at the different timepoints.

	Plasma Concentration Sorafenib ng/mL	Plasma Concentration Regorafenib ng/mL
5 min	104.0 (103.9)	20.2 (13.1)
20 min	80.0 (53.9)	105.9 (176.8)
1 h	139.7 (107.3)	23.4 (19.0)
24 h	63.5 (37.7)	54.7 (0.7)
72 h	13.2 (8.6)	47.2 (64.0)

Average volume of the bland beads was 0.75 mL.

**Table 3 cancers-15-03236-t003:** Average and standard deviation of serum concentration and embolized liver as well as non-embolized liver homogenate of regorafenib concentration.

	Regorafenib Plasma Concentration ng/mL	*p*-Value of Embolized Area vs. Plasma	Regorafenib Liver Homogenate of Embolized Area ng/mL	*p*-Value Embolized vs. Non-Embolized Area	Regorafenib Liver Homogenate of Non-Embolized Area ng/mL	*p*-Value of Non-Embolized Area vs. Plasma
Time of Sacrifice	50.5 (31.2)	0.05	8182.0 (8658.0)	0.03	408.4 (188.4)	0.003

## Data Availability

Data presented in this study are available upon request from the corresponding author. Some of the data are part of the patent submission.

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
