# Peer review of "Drug-Eluting Embolic Loaded with Tyrosine Kinase Inhibitor Targeted Therapies for Transarterial Chemoembolization in a VX2 Model"

_cancers, 2023, doi:10.3390/cancers15123236_

Round 1

Reviewer 1 Report

The topic of this manuscript is important and results could be interesting for readers. The work is well planned and written with the support of current methodology. However, some changes have to be entered into the revised version of the manuscript before it can be further processed:

1. In the experimental section, authors should provide more information about the preparation and characterization of the components of your material. The work is lack of materials views. The characterization should include SEM, drug release et al.

2. In the introduction section, the using of other microspheres should be discussed by citing and comparing with the paper (Development of PVA-based microsphere as a potential embolization agent).

3. Scale bars in figure 2 (100x magnification of H&E stain showing coagulative necrosis of the tumor associated 227 with beads seen in the arterial lumen.) and figure 3 (25x magnification of H&E stain demonstrates a clump of DEE necrotic area of liver with 230 massive lytic and coagulative necrosis of neoplastic parenchyma and adjacent hepatic parenchyma.), rather than simply provide the magnification in the figure captions. 

No further comments.

Author Response

  1. Thank you so much for your valuable comments

The topic of this manuscript is important and results could be interesting for readers. The work is well planned and written with the support of current methodology. However, some changes have to be entered into the revised version of the manuscript before it can be further processed:
Thank you so much for the opportunity and your valuable comments. 

1.    In the experimental section, authors should provide more information about the preparation and characterization of the components of your material. The work is lack of materials views. The characterization should include SEM, drug release et al.
Thank you for your comment. Some information was added however an entire paper can be written on that alone and the details beyond the scope of the paper however a description was added. The loading range was provided. Figures showing the elution profiles were added 
Briefly the tyrosine kinase inhibitor such as sorafenib para-toluene sulfonate salt, 1,2-dioleoyl-3- trimethylammonium-propane chloride salt (DOTAP) and 1,2-dioleoyl-sn- glycero-3-phosphocholine (DOPC) were dissolved in dehydrated ethanol. That solution was added dropwise to a 10% trehalose solution while stirring. The resulting emulsion was stirred at room temperature for an additional 5 minutes and filtered through a manual liposomal extruder using a 200 nm polycarbonate membrane. The uni-lamellar liposomal solution was lyophilized overnight. The liposomal formulated tyrosine kinase inhibitor was mixed with distilled water. Drug eluting embolics were drained out of the fluid leaving 2mL of beads which were added to the liposomal formulated sorafenib or regorafenib. The loading and elution profiles were examined in vitro. Although beyond the scope of the manuscript, the range of loading efficiency depending on conditions tested varied between 41- 51.9%. The elution profiles are shown in Figures 1 and 2. At 320 minutes, 82.6% of sorafenib and 74.2% of regorafenib had eluted off the beads. 

2.    In the introduction section, the using of other microspheres should be discussed by citing and comparing with the paper (Development of PVA-based microsphere as a potential embolization agent).
Thank you for this valuable reference. The paper was added in the introduction. “Embolic agents are being developed such as PVA-based imageable microspheres with improved tissue adhesion and loading capability of water-soluble anionic drugs such as 5-fluorouracil, but they cannot be loaded with lipophilic drugs.” 

3.    Scale bars in figure 2 (100x magnification of H&E stain showing coagulative necrosis of the tumor associated 227 with beads seen in the arterial lumen.) and figure 3 (25x magnification of H&E stain demonstrates a clump of DEE necrotic area of liver with 230 massive lytic and coagulative necrosis of neoplastic parenchyma and adjacent hepatic parenchyma.), rather than simply provide the magnification in the figure captions. 
Thank you, this will be done. The pictures were analysed by an independent professional veterinary pathology lab. We have sent them for the addition of a scale bar as requested. This will be forwarded as soon as it is received. We have submitted the revised paper to adhere to the deadline provided for the revisions.   

Reviewer 2 Report

General comments: the authors tried to prove that systematic agent "sorafenib" regorafenib" can be loaded in DEB in this rabbit experimental study. The topic is definitely of interest to readers. Overall, this manuscript was well written. 

Comments:

1. First paragraph of Discussion seems to be very lengthy, mostly background story. Please tried to shorten the "background study", please remove the statements that were not significantly related to your topics. 

2. So do you believe that sorafenib-loaded DEB TACE can be effective to treat HCC? Further can replace current method (Doxorubicin-loaded TACE)? This reviewer want to hear about the authors' opinion about this. 

3. Very recently, the paradigm of treatment for intermediate-stage HCC (BCLC B stage) is changing; the combination of TACE plus immunotherapy. Thus, many clinical trials about this combination therapy are ongoing. Please briefly mention and relate it with your study. 

Author Response

General comments: the authors tried to prove that systematic agent "sorafenib" regorafenib" can be loaded in DEB in this rabbit experimental study. The topic is definitely of interest to readers. Overall, this manuscript was well written. 
    Thank you so much, we appreciate your comments. 

Comments:
1.    First paragraph of Discussion seems to be very lengthy, mostly background story. Please tried to shorten the "background study", please remove the statements that were not significantly related to your topics. 
Thank you, we have removed statements and cut down from that paragraph however some of the background is essential in understanding the rationale behind the strategy of loading TKI on beads for local delivery during TACE. 

2.    So do you believe that sorafenib-loaded DEB TACE can be effective to treat HCC? Further can replace current method (Doxorubicin-loaded TACE)? This reviewer want to hear about the authors' opinion about this. 
This is the ultimate plan but further translational studies in large animal and clinical studies are needed. Yes in our opinion, sorafenib or Lenvatinib DEE-TACE should replace Dox-TACE because doxorubicin is not effective against HCC nor synergistic with TACE while tyrosine kinase inhibitors are. Future steps would involve comparison trials with doxorubicin loaded beads vs sorafenib loaded beads initially in animal and then humans. “Considering the lack of efficacy of doxorubicin loaded DEE-TACE and the systemic efficacy of tyrosine kinase inhibitors in HCC, sorafenib loaded DEE-TACE can have the potential to improve patient outcomes.”

3. Very recently, the paradigm of treatment for intermediate-stage HCC (BCLC B stage) is changing; the combination of TACE plus immunotherapy. Thus, many clinical trials about this combination therapy are ongoing. Please briefly mention and relate it with your study. 
The reviewer is correct. In fact, some large trials are combining TACE with systemic tyrosine kinase inhibitor (TKI) and 2 immunotherapies (NCT05301842, NCT05717738, NCT04814043, NCT04472767, NCT04803994). However, the more drugs’ patients have the more side effects and the lower the compliance. Therefore, loading TKI on DEE-TACE is very appealing. This was added to the discussion.